# Direct Interaction of Avian Cryptochrome 4 with a Cone Specific G-Protein

**DOI:** 10.3390/cells11132043

**Published:** 2022-06-27

**Authors:** Katharina Görtemaker, Chad Yee, Rabea Bartölke, Heide Behrmann, Jan-Oliver Voß, Jessica Schmidt, Jingjing Xu, Vita Solovyeva, Bo Leberecht, Elmar Behrmann, Henrik Mouritsen, Karl-Wilhelm Koch

**Affiliations:** 1Division of Biochemistry, Department of Neuroscience, University of Oldenburg, D-26111 Oldenburg, Germany; katharina.goertemaker@uni-oldenburg.de (K.G.); chad.yee@uni-oldenburg.de (C.Y.); janolivervoss1992@gmail.com (J.-O.V.); 2Animal Navigation, Institute of Biology and Environmental Sciences, University of Oldenburg, D-26111 Oldenburg, Germany; rabea.bartoelke@uni-oldenburg.de (R.B.); jessica.schmidt@uni-oldenburg.de (J.S.); jingjing.xu@uni-oldenburg.de (J.X.); bo.leberecht@uni-oldenburg.de (B.L.); henrik.mouritsen@uni-oldenburg.de (H.M.); 3Institute of Biochemistry, Faculty of Mathematics and Natural Sciences, University of Cologne, D-50674 Cologne, Germany; heide.behrmann@uni-koeln.de (H.B.); elmar.behrmann@uni-koeln.de (E.B.); 4Institute of Physics, University of Oldenburg, D-26111 Oldenburg, Germany; vita.solovyeva@uni-oldenburg.de; 5Research Center for Neurosensory Sciences, University of Oldenburg, D-26111 Oldenburg, Germany

**Keywords:** magnetoreception, cryptochrome, G protein α-subunit, protein-protein interaction

## Abstract

Background: Night-migratory birds sense the Earth’s magnetic field by an unknown molecular mechanism. Theoretical and experimental evidence support the hypothesis that the light-induced formation of a radical-pair in European robin cryptochrome 4a (ErCry4a) is the primary signaling step in the retina of the bird. In the present work, we investigated a possible route of cryptochrome signaling involving the α-subunit of the cone-secific heterotrimeric G protein from European robin. Methods: Protein–protein interaction studies include surface plasmon resonance, pulldown affinity binding and Förster resonance energy transfer. Results: Surface plasmon resonance studies showed direct interaction, revealing high to moderate affinity for binding of non-myristoylated and myristoylated G protein to ErCry4a, respectively. Pulldown affinity experiments confirmed this complex formation in solution. We validated these in vitro data by monitoring the interaction between ErCry4a and G protein in a transiently transfected neuroretinal cell line using Förster resonance energy transfer. Conclusions: Our results suggest that ErCry4a and the G protein also interact in living cells and might constitute the first biochemical signaling step in radical-pair-based magnetoreception.

## 1. Introduction

Sensing the Earth´s magnetic field is a widespread ability in the animal kingdom. Behavioral experiments and analyses of data from wild animals have shown that night-migratory songbirds, fish, amphibians, reptiles and insects orientate and navigate using magnetic sensing [1,2,3,4,5]. Birds sense the inclination of the magnetic field by a process that involves the absorbance of blue light, indicating the involvement of a photoreceptor protein [2,6,7,8]. In a seminal hypothesis, Schulten et al. (1978) proposed a radical-pair mechanism as the underlying mechanism for magnetic sensing [9], and a ubiquitously present class of chromophore proteins called cryptochromes (Cry) seemed to match the required properties for a magnetosensor [6,10,11]. Further research showed that magnetic compass information is detected in the retina and processed in the birds’ visual system [12]. However, the exact molecular basis of this complex process is currently unknown. 

Cry proteins are flavoproteins harboring a flavin adenine dinucleotide (FAD) in a binding pocket situated in a conserved photolyase homology region. The FAD chromophore enables Cry to absorb blue light, which triggers the formation of a radical pair involving a series of three or four neighboring tryptophan (Trp) residues [8,13]. 

Thus far, six different Crys (Cry1a, Cry1b, Cry2a, Cry2b, Cry4a and Cry4b) have been identified to be expressed in the retina of different bird species, but only Cry1a and Cry4a are currently discussed as the primary magnetic field receptors. Cry1a is localized in UV-sensitive cones in the retina of European robins [14] and a light-dependent immunostaining pattern of Cry1a in UV cones was interpreted to detect a light-triggered conformational change in Cry1a [14,15]. However, this light-dependent staining pattern of Cry1a in UV cones was not supported by a more recent study [16]. Instead, several lines of evidence point towards Cry4a being the magnetoreceptive protein. European robin Cry4a (*Er*Cry4a) can be purified with a bound FAD chromophore, exhibits a photo-induced electron transfer pathway leading to the formation of a [FAD^●−^ TrpH^●+^] radical pair, and it is magnetically sensitive in vitro [17,18,19]. Furthermore, Xu et al. (2021) found a substantially higher magnetic field effect in *Er*Cry4a than in chicken and pigeon Cry4 [19]. In a separate study, Hochstoeger et al. (2020) presented evidence for pigeon Cry4 acting as an ultraviolet-blue photoreceptor that forms photo-induced radical pairs [20]. These authors localized Cry4 in horizontal cells of the Pigeon retina and proposed a magnetic sensing mechanism by modulation of glutamatergic synapses. The immunohistochemistry of Cry4 in the Pigeon retina differs from results obtained with retinae from European robin, where *Er*Cry4 expresses in the outer segments of double cones and long-wavelength single cones but not in other retinal cells [21]. In the same study, Günther et al. (2018) co-localized *Er*Cry4 and iodopsin (long wavelength opsin) in long-wavelength single cones and in double cones [21].

A further important step for validating that Cry4 variants indeed operate as magnetoreceptive molecules is to link them to signal transduction processes causing magnetic-field-sensitive changes in the membrane potential. Wu et al. (2020) identified six putative protein interaction partners of *Er*Cry4a by a yeast two-hybrid screening approach [22]. All six genes code for retina-specific proteins. These proteins are *GNAT2* coding for the α-subunit of a cone-specific heterotrimeric G protein, *GNG10* coding for the γ-subunit of a cone-specific heterotrimeric G protein, *LWS*, also called iodopsin, coding for long-wavelength-sensitive opsin, *KCNV2* coding for potassium voltage-gated channel subfamily V member 2, *RBP1* coding for retinol-binding protein 1 and *RGR* coding for retinal G protein-coupled receptor. The identified α- and γ-subunit of the G protein are the cone-specific orthologues of the heterotrimeric G protein transducin (G_t_) that mediates phototransduction in mammalian and other vertebrate rod and cone photoreceptor cells. The versatile role of heterotrimeric G proteins in signal transduction pathways makes them primary candidates to test for direct interaction with Cry4 variants. Could the G protein be the long-sought-after first interaction partner of Cry4a in a radical-pair-based magnetoreception signaling pathway?

The aim of the present study is to test whether *Er*Cry4a directly interacts with the G protein α-subunit of European robin (*Er*G_t_α) on the molecular level. We expressed recombinant variants of both proteins, purified them and verified their functional state by independent functional assays. Using a biosensor approach employing surface plasmon resonance (SPR), we investigated the protein–protein interaction process and analyzed the kinetic parameter of the binding process, which was further corroborated by in vitro pulldown affinity assays. Using acceptor photobleaching Förster resonance energy transfer (FRET), we further validated that the interaction between *Er*Cry4a and *Er*G_t_α also occurs in living cells in a neuroretinal bird cell line.

## 2. Materials and Methods

### 2.1. General Cloning Strategies

All primers for cloning steps are listed in the Appendix A. The coding sequence of *Er*Gnat2 (coding for cone-specific Gα from European robin; see Appendix A for sequence information and alignments) was present in the pGBKT7 vector as described by Wu et al. [22]. A PCR was performed to amplify the coding sequence and add flanking *Spe*I and *Xho*I restriction sites. The product was subsequently digested and ligated with digested pCold vector (Takara Bio, Shiga, Japan) to create the pCold *Er*Gnat2 construct. 

To create the chimera, a deletion mutagenesis of amino acids 220–298 of *Er*Gnat2 in the pCold vector was performed, and the corresponding region of bovine inhibitory alpha subunit was amplified from the commercial pCS6 vector containing bGNA1i. The products were then assembled using the Gibson assembly to create the completed pCold *Er*Gnat2 chimera. To create the pET21a *Er*Gnat2 chimera vector, the chimera sequence was amplified from pCold and assembled using the Gibson assembly with NdeI and XhoI digested pET21a. To create the pACE SUMO *Er*Gnat2 chimera vector, the chimera sequence was amplified and combined with the pACE SUMO vector using the Sequence and Ligation Independent Cloning (SLIC) technique. To remove 6 non-native amino acids from the coding sequence, PCR mutagenesis was performed. 

### 2.2. Cloning of the G_t_α/Giα Chimera Containing a SUMO Tag

Earlier work on mammalian G_t_α showed that expression in *E. coli* resulted in misfolded protein [23], and we obtained the same result by trying to express wildtype *Er*G_t_α. However, it is common practice to construct, express and purify chimeric mammalian G_t_α proteins for functional studies [23]. The chimeric G-protein α-subunit G_t_α/Giα is based on the coding sequence for the *European robin* gene *ErGNAT2* [22] encoding *Er*Gtα and the *Bovine taurus* Giα1 (Appendix A). The sequence was cloned seamlessly into a T7lac expression vector containing a T7lac promoter and a Small Ubiquitin-Related Modifier (SUMO) protein tag modified with an N-terminal 6x histidine tag (Appendix A). The initiator methionine of the *ErGNAT2* sequence was removed, and plasmids were confirmed by Sanger sequencing (Eurofins Genomics). Primer pairs utilized and more details about the cloning steps can be found in Appendix A. 

The plasmid was transformed into chemically competent BL21 *E. coli* cells (prepared in-house). Subsequently, the cells were cultured at 37 °C and 180 rpm in LB medium containing 10 g/L NaCl and 100 µg/L ampicillin (Carl Roth, Karlsruhe, Germany). Upon reaching an OD_600_ of 0.6, the shaking was reduced to 160 rpm and the cells were cooled to 17 °C for 30 min. Next, the cells were induced by adding IPTG (Roth) to 10 µM, incubated for 20–24 h at 17 °C, and then harvested and stored at −20 °C. 

Cells from 3.5 L culture were resuspended in 50 mL Ni-NTA binding buffer (20 mM HEPES pH 7.8, 250 mM NaCl, 10 mM MgCl_2_, 10 mM Imidazole, 10 mM ß-mercaptoethanol, Roche cOmplete™ EDTA-free Protease Inhibitor Cocktail) and lysed by sonication using a Bandelin GM 2200 ultrasonic generator with a UW 2200 converter and MS72 microtip at 45% power. Cell lysates were then centrifuged at 18,000 rpm (~25,000 g avg) in a JA25.5 rotor at 4 °C for 60 min, and the supernatants were applied to equilibrated gravity flow columns containing 2 mL of Ni-NTA resin each. Each column was washed with 80 matrix volumes (160 mL) wash buffer (10 mM HEPES pH 7.8, 250 mM NaCl, 10 mM MgCl_2_, 20 mM imidazole, 10 mM β-mercaptoethanol), and then eluted twice using 4 mL of elution buffer (20 mM HEPES pH 7.8, 150 mM NaCl, 10 mM MgCl_2_, 300 mM imidazole, 10 mM β-mercaptoethanol). 

Both elution fractions were dialyzed together (SERVAPOR^®^ dialysis tubing, MWCO 12,000–14,000 RC, diameter 21 mm) after adding Ulp1 protease (in-house) against at least 4 L of dialysis buffer (10 mM HEPES pH 7.8, 250 mM NaCl, 10 mM MgCl_2_) overnight. The digest was then applied to the same equilibrated gravity flow columns containing Ni-NTA matrix to remove digested tag sequences. For this, the columns were washed 3 times with 5 mL wash buffer and eluted with 5 mL elution buffer. The flow-through and wash fractions containing the Gtα/Giα chimera were pooled and concentrated using a Macrosep Advance 10k MWCO centrifugal filter to a concentration of up to 15 mg/mL. Concentrated fractions were applied to house-packed size exclusion chromatography column (Superdex75, volume of 320 mL) pre-equilibrated with size exclusion chromatography buffer (10 mM HEPES pH 7.4, 150 mM NaCl, 10 mM MgCl_2_, 3.4 mM EDTA). The flow rate was maintained at 1 mL/min, and fractions containing the G-protein were identified with SDS Page, pooled, and concentrated and, then, flash frozen in liquid N_2_ in 50 or 100 µL aliquots and stored at −80 °C. 

### 2.3. Expression and Purification of the Myristoylated G_t_α/G_i_α Chimera

*E. coli* BL21 (Codon+) cells, harboring the pET21a G_t_α/G_i_α chimera plasmid and pBB131 encoding yeast myristoyltransferase, were grown in 5 × 500 mL yeast tryptone (YT) media at 37 °C and 180 rpm in the presence of 100 µg/mL ampicillin (Roth) and 30 µg/mL kanamycin (Carl Roth, Karlsruhe, Germany). At OD_600_ of 0.4 myristic acid (Fluka, Buchs, Switzerland) was added to a final concentration of 50 µg/mL, and the incubation was continued until an OD_600_ of 0.5–0.6 was reached. Then, the cultures were cooled down to 17 °C at 160 rpm for 30 min. IPTG (Roth) was added to a final concentration of 150 µM, and incubation was continued for 20–24 h at 17 °C and 160 rpm. Afterward, the cell pellets were harvested by centrifugation at 7000 rpm for 7 min at 4 °C. Each cell pellet was resuspended in 10 mL in Ni-NTA binding buffer (20 mM Tris-HCl pH 8.0, 250 mM NaCl, 10 mM imidazole) and a protease inhibitor cocktail (Roche cOmplete™ EDTA-free Protease Inhibitor Cocktail) was added (one tablet dissolved in 2 mL of water, of which 0.2 mL were added to 10 mL of Ni-NTA binding buffer). The cells were lysed by ultrasonication using a Bandelin GM 2200 ultrasonic generator with a UW 2200 converter and MS72 microtip at 45% power and centrifuged at 100,000× *g* for 30 min at 4 °C. The supernatant was used for further purification. An empty 20 mL column was packed with 2.5 mL Ni^2+^-nitrilotriacetic acid-agarose resin (HisPurTM Ni-NTA) and equilibrated with 2 × column volumes of Ni-NTA binding buffer. The supernatant was loaded on the Ni-NTA column by gravity flow and washed with 50 × matrix volume of Ni-NTA washing buffer (20 mM Tris-HCl pH 8.0, 150 mM NaCl, 20 mM imidazole, 10 mM β-mercaptoethanol). The protein was eluted with 2 × matrix volume of Ni-NTA elution buffer (20 mM Tris-HCl pH 8.0, 150 mM NaCl, 300 mM imidazole, 10 mM β-mercaptoethanol). Next, an anion exchange chromatography (AEC) was performed. The elution fraction from the NTA-affinity chromatography was diluted 1:3 in AEC buffer A (20 mM Tris-HCl pH 7.0 and 1 mM DTT). A HiTrap^TM^ 5 mL Q Sepharose High Performance (QHP, Cytiva, Uppsala, Sweden) column was equilibrated with 5% AEC buffer B (20 mM Tris-HCl pH 7.0, 1 M NaCl and 1 mM DTT) and 95% AEC buffer A. The protein sample was loaded on the column and washed with 5% AEC buffer B and a flow rate of 1 mL/min until a stable UV signal was reached. The protein was eluted with a salt gradient from 5 to 50% AEC buffer B over 112.5 mL at a flow rate of 1 mL/min. Fractions containing the G_t_α/G_i_α chimera were pooled, and to remove all remaining contamination, a size exclusion chromatography (SEC) was performed. A Superdex 200 16/600 column with a volume of 120 mL was equilibrated using SPR buffer (10 mM Hepes pH 7.4, 150 mM NaCl, 10 mM MgCl_2_, 3.4 mM EDTA), and the pooled AEC fractions were concentrated using a Macrosep Advance 10k MWCO centrifugal filter. Using a flow rate of 0.5 mL/min, the sample was applied to the column, and the SEC run started. Fractionation was started after 0.4 CV using 3 mL fraction sizes. Desired fractions were pooled, concentrated, and flash frozen in liquid N_2_ for storage at −80 °C. Myristoylation was analyzed and verified by reversed phase analytical HPLC as described previously for myristoylated neuronal calcium sensor proteins [24], yielding 80% myristoylation. SDS-PAGE analyses of Gtα/Giα chimera variants did not show any heterogeneity indicating higher myristoylation rates (Appendix A).

### 2.4. Cloning, Expression, and Purification of ErCry Variants

Basic steps of cloning, expression and purification of ErCy4a have been described previously [19]. The following modifications were applied: the LB media for expression contained 10 g/L yeast extract instead of 5 g/L. The C-terminally truncated mutant *Er*Cry4-497 was produced by mutagenesis using forward and reverse primers as listed in the Appendix A on the WT *Er*Cry4a pCold plasmid as described [19]. *Er*CRY4-497 was expressed and purified as *Er*Cry4a, except that the expression time was extended from 22 to 44 h. 

*Er*Cry1a and *Er*Cry1b, as well as the chimeric G-protein α-subunit Gtα/Giα were cloned into the pFastBacHT B vector (Thermo Fisher, Waltham, MA, USA) using *BamH*I and *Xho*I restriction sites and primers listed in Appendix A. Mouse-codon optimized *Er*CRY4 in the pFast vector was a gift from Joseph S. Takahashi (University of Texas Southwestern Medical Center, Dallas, TX, USA).

Baculovirus was produced in SF9 cells (Thermo Fisher) using the Bac-to-Bac Baculovirus expression system (ThermoFisher), while Tni cells (BioTrend, Cologne, Germany) were used for protein expression. Cells from 1 L culture were sedimented by centrifugation at 3745× *g* and resuspended in 30 mL homogenization buffer (50 mM Tris, pH 8.8, 300 mM NaCl, 15 mM imidazole, 10 mM β-mercaptoethanol, Roche cOmplete™ EDTA-free Protease Inhibitor Cocktail) per 10 g cell pellet. Cells were lysed with a Potter-Elvehjem-homogenizer and clarified by centrifugation at 48,384× *g*. Clarified cell lysates were applied to Ni-NTA agarose columns (Qiagen, Hilden, Germany), pre-equilibrated with homogenization buffer. Bound proteins were eluted with elution buffer (50 mM Tris, 300 mM NaCl, 400 mM Imidazole and 10 mM β-mercaptoethanol; pH 8.0 for *Er*Cry4 and Gtα/Giα, and pH 8.8 for *Er*Cry1a and *Er*Cry1b). The proteins were diluted 1:10 in 20 mM Tris and further purified on anion-exchange 5 mL Hitrap Q columns (Cytiva, Uppsala, Sweden), after equilibration with buffer A (20 mM Tris, 30 mM NaCl, and 10 mM β-mercaptoethanol; pH 8 or 8.8). Proteins were eluted with a gradient increasing NaCl to 0.5 M at a flow rate of 1 mL/min. While fractions containing *Er*Cry4 were yellow in color, due to the absorption of bound FAD, those of *Er*Cry1a and *Er*Cry1b were colorless.

### 2.5. Limited Proteolysis of the G_t_α/G_i_α Chimera

The proteolytic digestion was performed under conditions such as those previously described [25,26] for tryptophan fluorescence analysis. Purified and frozen G_t_α/G_i_α chimera samples were thawed. Two reaction mixtures were prepared, both containing 50 mM Tris/HCl at pH 7.4, 50 mM NaCl, 2 mM MgCl_2_, 10 µM GDP, and 10 µM G_t_α/G_i_α chimera. To one reaction, 50 µM AlCl_3_ and 10 mM NaF were added. The reactions were started by adding chymotrypsin (Roth), from a frozen stock solution containing 0.5 mg/mL chymotrypsin in 0.5 mM HCl and 50% glycerol, to 12.5 µg/mL and a final volume of 300 µL. Upon addition of protease, the reactions were incubated at 37 °C. Digestion was stopped after 0.5, 2, 5, 10, 20, 40, 60, and 90 min by removing 30 µL of the reaction mixture each, mixing with 10 µL of 4× Laemmli sample buffer, heating to 95 °C for 3 min, and storage at −20 °C. Additionally, a sample was prepared before the addition of protease. Analysis was performed by loading identical volumes of each sample onto 10% Bis-tris SDS-PAGE gels. Gels were stained using Coomassie Brilliant Blue R250 and decolorized using 10% acetic acid and 40% ethanol. Visualization was performed using an Azure c400 Gel Imaging System by Azure Biosystems, Dublin, CA, USA.

### 2.6. Trp Fluorescence Emission

The intrinsic Trp fluorescence assay for measuring the activation-dependent conformational change in G protein α-subunits is a widely used assay to test their functional intactness. Details of the experimental setup are described [26]. We recorded relative fluorescence emission on a spectrofluorimeter from Photon Technology International. Purified G_t_α/Giα sample was thawed and diluted in fluorescence buffer (50 mM Tris pH 7.4, 50 mM NaCl, 10 µM GDP and varying concentrations of MgCl_2_) at a final concentration of 1 µM. Trp fluorescence emission was recorded at 340 nm after excitation at 290 nm. Changes in emission rate were triggered by injection of 50 µM AlF_4_^−^ (premix of 50 µM AlCl_3_ and 10 mM NaF). Recording time was 500 s.

### 2.7. Pulldown Experiments

*Purification of ErCry4 and G_t_α/G_i_α chimera*—Both *Er*Cry4a and non-myristoylated G_t_α/G_i_α chimera were expressed and purified essentially as described above, except that all steps were carried out under far-red light (750+ nm). For the G_t_α/G_i_α chimera, concentrated fractions after affinity purification were not digested to remove the tag but directly purified by size exclusion chromatography using a Superdex200 increase 10/300 column pre-equilibrated with a running buffer (20 mM HEPES, pH 7.4, 150 mM NaCl, 10 mM MgCl_2_). The flow rate was maintained at 0.17 mL/min, and fractions containing the G-protein were identified with SDS-PAGE and used immediately for pulldown experiments. For *Er*Cry4a, after affinity purification, elution fractions were dialyzed together (Spectra/Por dialysis tubing, MWCO 6000–8000) after adding TEV protease (in-house) to an approximate molar ratio of 1:20 against at least 3 L of dialysis buffer (20 mM HEPES, pH 7.4, 200 mM NaCl, 10 mM MgCl_2_) overnight. The digest was then applied to the gravity flow columns (Bio-Rad Laboratories GmbH, Feldkirchen, Germany containing Super Ni-NTA agarose resin (Anatrace) equilibrated with dialysis buffer to remove His-tag and protease. The flow-through containing *Er*Cry4a was concentrated using a Vivaspin Turbo 15 10k MWCO centrifugal filter and further purified by size exclusion chromatography using a Superdex200 increase 10/300 column (Cytiva, Uppsala, Sweden) pre-equilibrated with a running buffer (20 mM HEPES, pH 7.4, 150 mM NaCl, 10 mM MgCl_2_). The flow rate was maintained at 0.17 mL/min, and fractions containing *Er*Cry4a were identified with SDS-PAGE and used immediately for pulldown experiments.

*Pulldown*—All steps until SDS-PAGE gel electrophoresis were carried out under far-red light (750+ nm). For protein pulldown experiments, 1.47 nmol each of *Er*Cry4a and His-G_t_α/Giα were mixed and incubated overnight at 6 °C with gentle agitation. For the negative controls, His-G_t_α/Giα was replaced with a running buffer. Next, 50 µL Super Ni-NTA agarose resin (Anatrace, Maumee, OH, USA) was added, and the sample was incubated for an additional 1.5 h at 6 °C with gentle agitation. The slurry was transferred to Micro Bio-SpinTM Columns (Bio-Rad Laboratories GmbH, Feldkirchen, Germany), washed five times with 1 mL of wash buffer (20 mM HEPES, pH 7.4, 150 mM NaCl, 10 mM MgCl_2_, 20 mM imidazole), then incubated with 70 µL elution buffer (20 mM HEPES, pH 7.4, 150 mM NaCl, 10 mM MgCl_2_, 500 mM imidazole) for 30 min on ice. For elution, columns were centrifuged for 1 min at 4 °C and 500 g. SDS-PAGE samples were prepared as follows. For the input controls, 2.5 µg of either *Er*Cry4a or His-G_t_α/Giα were mixed with SDS-sample buffer, boiled for 5 min at 95 °C, and then placed on ice until loaded on the gel. For the other samples, 20 µL of the elution fraction were mixed with 5 µL SDS-sample buffer, boiled for 5 min at 95 °C, then placed on ice until loaded on the gel. Samples were run on 4–20% Tris SDS-PAGE gradient gels. Gels were stained using Coomassie Brilliant Blue R250 (Sigma-Aldrich, Taufkirchen, Germany and decolorized using 10% acetic acid and 40% ethanol. Visualization was performed using a Fusion FX6 Edge (Vilber, Eberhardzell, Germany) imaging system.

*Densiometric analysis*—Analysis was carried out in the Fiji software package using the SDS-PAGE analysis function allowing for the integration of grey values with background correction. For each experiment, the *Er*Cry4a input band was quantified. Then the same region of interest was quantified for negative control and pulldown lanes. The bar diagram visualizes the average percentage of input *Er*Cry4a recovered in negative control and pulldown, respectively, with error bars indicating the standard deviation. Individual experimental results are indicated by circles.

### 2.8. Surface Plasmon Resonance

Surface plasmon resonance (SPR) measurements were performed on a Biacore 3000 (GE Healthcare now Cytiva). The general operation principle, including the immobilization procedures and quantitative data analysis, has been described before [27,28]. We used CM5 sensor chips (GE Healthcare) for all applications. The carboxy-methyl dextran coated sensor chip surface of CM5 sensor chips was activated by carbodiimide/N-hydroxy-succinimide chemistry (Biacore Immobilization Kit, Cytiva, Uppsala, Sweden, allowing subsequent covalent coupling of proteins via free NH_2_-groups. In preliminary tests, we compared different immobilization geometries, immobilization densities and regeneration protocols. Immobilization densities of myristoylated Gtα/Giα were 3.6–3.7 ng/mm^2^ and of non-myristoylated Gtα/Giα were 2.6–3.9 ng/mm^2^. Interaction processes were studied by injection of *Er*Cry variants at different concentrations at a flow rate of either 5 or 20 µL/min. Some recordings were also performed at higher flow rates of 30 and 50 µL/min. SPR running buffer was 10 mM HEPES/NaOH, pH 7.4, 150 mM NaCl, 10 mM MgCl_2_, 0.005% Tween-20, 3.4 mM EDTA. Control surfaces were coated with Ulp1 (in-house made) at a density of 2.4–3.6 ng/mm^2^ or by an amino coupling activation/deactivation cycle (SPR recordings with *Er*Cry1a). 

For the evaluation of sensorgrams, we used nonlinear curve fitting implemented in the BIAevaluation software 4.1 (GE Healthcare, Boston, MA, USA) by applying the global fitting approach. Sensorgrams obeying a mono-exponential Langmuir binding process (A + B ↔ AB) yielded association and dissociation rate constants and apparent K_D_ values. SPR sensorgrams that did not show mono-exponential binding curves were evaluated by a two-state-reaction model according to A + B ↔ AB ↔AB* (* indicates a different protein conformation). The latter describes the binding process of A + B by a forward rate constant k_a1_ and a backward rate constant k_d1_ leading to complex formation AB and a conformational change of AB to AB* (BIAevaluation software 4.1). We calculated the apparent K_D_ from the ratio of k_d1_/k_a1_.

### 2.9. FRET Measurements

*Cloning of FRET constructs*—cDNA production: RNA was extracted from the retina of one European robin, which was wild-caught in the vicinity of the university campus using mist nets. The animal was sacrificed by decapitation, and the eyes were immediately removed. The retina, free of vitreous, was shock-frozen in liquid nitrogen and stored at −80 °C until RNA extraction. RNA was extracted using the NucleoSpin RNA XS kit (Macherey Nagel, Düren, Germany), and a cDNA library was generated using the Make Your Own “Mate & Plate^TM^” Library System (Takara Bio, Shiga, Japan).

All FRET constructs were based on the pKan-CMV-mClover3-mRuby3 vector, a gift from Michael Lin (Addgene plasmid #74252; http://n2t.net/addgene (accessed on 26 February 2020): 74252; RRID: Addgene_74252) [29]. Table 1 gives an overview of FRET plasmids and expressed proteins. pKan-CMV-mClover3-*Er*CRY4 was generated by first linearizing pKanCMV-mClover3-mRuby3 with primers 1 and 2 (Appendix A) using PrimeSTAR Polymerase (Takara Bio, Shiga, Japan), and amplifying the *ErCRY*4 cDNA with primer 3, introducing a *BamH*I and *Asc*I restriction site and a Ser-Gly-Ser-Ser-Gly-Ser-Ser-Gly linker between mClover3 and *ErCRY*4, and the reverse primer 4, introducing a *Xho*I restriction site and a stop codon after *ErCRY*4. The linearized vector and gene product were then recombined using In-Fusion (Takara Bio).

pKan-CMV-mClover3 was generated from pKan-CMV-mClover3-*ErCRY*4 by deletion of *ErCRY*4 and mRuby3 by using the Q5 site-directed mutagenesis kit (New England Biolabs) with primers 5 and 6, introducing a *Xho*I restriction site and a stop codon after the mClover3 plus linker. pKan-CMV-mClover3-*ErCRY*4-497 was generated by amplifying the *ErCRY*4 cDNA with primers 7 and 8 and In-Fusion recombination with pKan-CMV-mClover3 digested with *Asc*I and *Xho*I. pKan-CMV-mRuby3 was generated by deletion of mClover3 from pKan-CMV-mClover3-mRuby3 using the Q5 site-directed mutagenesis kit (New England Biolabs, Ipswich, MA, USA) and primers 9 and 10. pKan-CMV-*ErGNAT2*-mRuby3, with *GNAT2* being the corresponding gene to the wildtype *Er*G_t_α protein, was generated by first linearizing pKan-CMV-mClover3-mRuby3 with primers 11 and 12 using PrimeSTAR Polymerase, and then amplifying the *GNAT2* gene with primers 13, introducing an *Asc*I restriction site, and 14, introducing a *Xho*I restriction site. Both PCR products were then recombined using In-Fusion, resulting in the deletion of mClover3 and a fusion construct of *GNAT2*-mRuby3. The *GNAT*2 sequence was compared to the genome sequence (NCBI Sequence ID LR812129.1).

*Cell Culture and Expression*—The QNR/K2 Neuroretina Quail cell line was purchased from ATCC (CRL-2533) and cultured in DMEM + GlutaMAX (Gibco, Waltham, MA, USA) supplemented with 10% fetal bovine serum (Gibco) at 39 °C and 5% CO_2._ QNR/K2 cells were (co-)transfected with the FRET constructs using Lipofectamine 2000 (Invitrogen) according to manufacturer’s instructions, after which cells were handled in dim red light until fixed. Forty-eight hours after transfection, cells were washed twice with PBS (Gibco), fixed in 4% (wt/v) paraformaldehyde for 30 min, and washed again twice with PBS and kept in PBS for imaging. Transfection, fixation and imaging were all performed in µ-Slides 8 Well with a #1.5 polymer coverslip bottom (Ibidi). 

*Acceptor photobleaching FRET*—Imaging of transiently transfected QNR/K2 cells was performed with an inverted Leica TCS SP5 II confocal microscope. A 63×/water objective with 1.2 numerical aperture was used (PL APO, corr CS). A white light laser was used as an excitation light source. The excitation wavelengths were set to 488 nm for mClover3 fluorescent protein and 560 nm for mRuby3. Photomultiplier tubes were used as detectors. The detection range was set to 500–550 nm for mClover3 emission and 570–644 nm for mRuby3 emission using AOTF (acousto-optical tunable filter). To determine the FRET efficiency, the acceptor photobleaching protocol was used. Acceptor photobleaching was performed with the FRET AB wizard in the LAS AF software on selected regions of interest (ROIs), generally one to two per cell in a region where both fluorophores showed expression. Acceptor bleaching was achieved by setting four parallel laser lines of white light laser (560, 568, 576, and 584 nm) to 100% intensity. Donor bleaching was found to be negligible under these conditions. Prebleach and postbleach images were acquired. FRET for the ROIs was observed by an increase in donor (mClover3) fluorescence intensity following the acceptor (mRuby3) photobleaching. FRET efficiency was measured in percent and calculated automatically by Leica LAS AF software as (D_post_ − D_pre_)/D_post_, where D_post_ (D_pre_) is the fluorescence intensity of the donor after (before) photobleaching. For each condition, at least three individual experiments from freshly transfected cells were performed.

*Statistics*—Since the FRET efficiency read-outs were reported as percentages, we opted for a binomial generalized linear model (GLM) followed by an analysis of variance, type III from the ‘car’ package in reference [30] of the resulting models to compare interactions and negative controls. The data were analyzed with a custom-written R-script [31]. 

## 3. Results

### 3.1. Interaction of ErCry4a with the G Protein α-Subunit of European Robin (G_t_α/Giα Chimera)

For interaction studies with *Er*G_t_α, we used purified recombinant *Er*Cry4a that matches the in vitro requirements for a magnetic sensing protein [19]. However, heterotrimeric G proteins from European robin have not been cloned, heterologously expressed or purified so far. To obtain soluble *Er*G_t_α, we constructed a G_t_α/Giα chimera, in which a region of 79 amino acids was replaced by the corresponding region from an inhibitory bovine Gα-subunit. The inserted region is highly homologous to the *Er*G_t_α counterpart and differs only in 23 positions (of which ten are conservative replacements). The benefit is that the replacements facilitate heterologous expression and subsequent purification as a soluble protein [23] (see Appendix A). Thus, the G_t_α/Giα chimera was used as a surrogate for G_t_α.

Using surface plasmon resonance (SPR), we investigated the binding affinities and kinetic parameters of the interaction between *Er*Cry4a and the Gtα/Giα chimera. We took into account that α-subunits of G proteins are myristoylated at the amino-terminus and expressed and purified myristoylated and non-myristoylated G_t_α/Giα chimera. Both variants were immobilized by a standard SPR amine coupling procedure on a CM5 sensor chip surface [27]. In a second step, we supplied purified *Er*Cry4a in the mobile phase that is flushed over the G_t_α/Giα chimera coated chip surface. A control surface was coated with ubiquitin-like-protease 1 (Ulp1). Purified *Er*Cry4a was injected and flushed over both surfaces (control and the G_t_α/Giα chimera coated surface), and the control recordings were subtracted from the sample recordings. A positive increase in SPR resonance units (RU) indicated a binding process and injection of increasing *Er*Cry4a concentrations resulted in sensorgrams of increasing amplitudes (black curves in Figure 1). Figure 1A shows the interaction of ErCry4a with myristoylated G_t_α/Giα. We performed a global fitting approach to obtain one value for the association and dissociation rate constant. A simple Langmuir binding model did not result in robust fits of sensorgrams. However, a global fitting approach was successful (Figure 1, red lines) when we applied a two-state-reaction model according to A + B ↔ AB ↔AB*, which takes into account a conformational change from AB to AB* (*indicates a different protein conformation). Global fitting using a two-state-reaction model (see Methods) of the association and dissociation phases (Figure 1A) resulted in an apparent K_D_ of 0.20 µM based on an association rate constant k_a1_ = 3.32 × 10^3^ M^−1^ s^−1^ and dissociation rate constant k_d1_= 6.74 × 10^−4^ s^−1^. Fitting three data sets, including a total of nine injections, yielded a K_D_ = 0.29 ± 0.08 µM (n = 3).

Binding of *Er*Cry4a to non-myristoylated G_t_α/Giα also occurred in a concentration-dependent manner (Figure 1B). By performing the same fitting routine (two-state-reaction model), we analysed five different data sets similar to the example shown in Figure 1B. Forward and backward reaction rates for the formation of AB gave an apparent K_D_ = 35 ± 15 nM (n = 5). These results showed a moderate to high affinity of *Er*Cry4 to myristoylated G_t_α/Giα and a high affinity to non-myristoylated G_t_α/Giα, indicating an impact of the myristoyl group for the interaction process (see Discussion for interpretation of K_D_ values).

The C-terminus of pigeon and chicken Cry4 seems to be involved in light–induced conformational changes [18,32,33]. We tested the binding of myristoylated G_t_α/Giα to a C-terminally truncated variant of *Er*Cry4a (*Er*Cry4a-497) and determined an apparent K_D_ of 0.2 ± 0.14 µM (n = 3, a representative example is the upper sensorgram in Figure 1C). A similar affinity with a K_D_ = 0.34 ± 0.12 µM was determined for *Er*Cry4a-497 and non-myristoylated G_t_α/Giα (Figure 1C, curve 2). 

### 3.2. Interaction of ErCry4a and G_t_α/G_i_α in Solution

SPR experiments require the immobilization of one binding partner on a sensor surface. To verify that ErCry4a and G_t_α/G_i_α also interact in solution, we performed pulldown experiments. For this, we pre-incubated a mixture of purified *Er*Cry4a and purified His-tagged G_t_α/Giα chimera to allow for a complex formation and then incubated the protein solution with a Ni-NTA affinity resin expected to bind to the His-tag present only on the G_t_α/Giα chimera. As a negative control, we omitted the His-tagged G_t_α/Giα chimera. After stringent washing, bound proteins were eluted from the resin and visualized using sodium dodecyl sulphate polyacrylamide gel electrophoresis (SDS-PAGE) analysis (Figure 2). This revealed that *Er*Cry4a was washed from the matrix in the absence of the G_t_α/Giα chimera, whereas it co-eluted, albeit at varying levels, when the G_t_α/Giα chimera was present.

### 3.3. Switch of G_t_α/Giα to the Active Conformation

We verified that both G_t_α/Giα variants purified by Ni-NTA affinity chromatography, anion exchange chromatography (AEC) and/or size exclusion chromatography (SEC) are functionally active by measuring their well-known conformational change in response to GDP/GTP exchange (Figure 3 and Appendix A). This conformational change is typical for G protein α-subunits and can be monitored by Trp fluorescence spectroscopy [34]. A unique Trp is conserved at or near position 207 in all α-subunits of heterotrimeric G proteins from vertebrates, and the corresponding position in *Er*G_t_α is 211. The addition of AlF_4_^−^ to a purified Gα-protein with bound Mg^2+^-GDP leads to a protein conformational change resembling the transition to the active state, which is measured by an increase in Trp fluorescence emission [25,35,36]. We elicited the increase in Trp fluorescence emission by adding AlF_4_^−^ and MgCl_2_. Figure 3 shows a concentration-dependent increase in Trp fluorescence emission, when AlF_4_^−^ was applied to G_t_α/Giα samples pre-incubated with different Mg^2+^-concentrations. Our results match previous recordings obtained with bovine rod outer segment G_t_α and indicate that our G_t_α/Giα chimera is an active protein [25,35] that might undergo a conformational change during the binding process (Figure 1). To corroborate the conformational changes in the presence and absence of AlF_4_^−^, we additionally carried out limited proteolysis of the Gtα/Giα chimera. The chimera G_t_α/G_i_α was less susceptible to proteolysis in the presence of AlF_4_^−^, indicating a switch to the active conformation (Appendix A), which is in agreement with previous reports about bovine rod outer segment transducin G_t_α [35,36].

### 3.4. Cellular FRET Analysis of the ErCry4a and ErG_t_α Interaction Process

Our SPR and pulldown tests demonstrated that *Er*Cry4a and G_t_α/Giα interact in vitro. Next, we wanted to verify that *Er*Cry4a interacts in a cellular environment with the wildtype form of *Er*G_t_α. This is important to exclude that the observed interactions were not compromised by the use of a chimeric construct. For this purpose, we transiently transfected a neuroretinal quail cell line with fluorescently labeled *Er*Cry4a and *Er*G_t_α to perform acceptor photobleaching FRET [37]. We employed mClover3 and mRuby3 as donor and acceptor fluorophores, respectively. We fused mClover3 to the N-terminus of *Er*Cry4a and *Er*Cry4a-497, and mRuby3 to the C-terminus of *Er*Gtα. When interacting, the donor fluorescence increases after bleaching the acceptor since fewer acceptor fluorophores are available for energy transfer. As a positive control, we used a fusion construct of mClover3 and mRuby3. In this construct (Figure 4, Clov3-Ruby3), donor and acceptor constitute a single protein being close enough for energy transfer to occur. Negative controls included the fusion constructs in the presence of free acceptor or free donor fluorophores (Figure 4, Clov3 + Gtα-Ruby3, Clov3-Cry4 + Ruby3 and Clov3-Cry497 + Ruby3). 

Indeed, we observed a significantly higher FRET signal for the combination of *Er*G_t_α-mRuby3 with mClover3-*Er*Cry4a or with mClover3-*Er*Cry4a-497 (Figure 4 and Table 2; mean ± s.d.: Clov3-Cry4 + Gtα-Ruby3 = 49.7 ± 10.9%; and Clov3-Cry4-497 + Gtα-Ruby3 = 49.5 ± 13.8%) than the negative controls mClover3 + Gtα-mRuby3 and mClover3-*Er*Cry4a or mClover3-*Er*Cry4a-497 with mRuby3 (Figure 4 and Table 2; mean ± s.d. Clov3+ Gtα-Ruby3 = 12.2 ± 10.4%; Clov3-ErCry4a + Ruby3 = 17.6% ± 9.6%; and Clov3-ErCry4a-497 + Ruby3 = 23.7% ± 10.6%). 

We observed a slightly higher false FRET background of mClover3-*Er*Cry4-497 + mRuby3 compared to the other negative controls and a wider spread of the data in both groups, including mClover3-*Er*Cry4-497, possibly due to differences in expression. Nonetheless, the mean value of the FRET efficiency of Gtα-mRuby3 with mClover3-*Er*Cry4 with and without the C-terminus was nearly identical (49.7% versus 49.5%), supporting that the C-terminus is not involved in the interaction. The acceptor photobleaching technique tells us that *Er*G_t_α and *Er*Cry4a and *Er*Cry4a-497 are in close proximity to each other, indicating a protein–protein interaction.

### 3.5. Interaction of ErCry1 Forms with G_t_α/Giα

Heterotrimeric G proteins can couple to a variety of different G protein-coupled receptors, making them suitable for multiple G protein-coupling and G protein targeting [38,39]. Since *Er*Cry1a is the other Cry form currently discussed as a potential magnetoreceptive protein [14,15], we included *Er*Cry1a and *Er*Cry1b in our interaction analysis by testing for binding of these purified *Er*Cry1 forms (the apo forms without any bound FAD) to myristoylated G_t_α/Giα by SPR. Using the same experimental approach as for *Er*Cry4a, *Er*Cry1a and *Er*Cry1b showed a concentration-dependent binding to myristoylated G_t_α/Giα (Figure 5). A two-state-reaction model revealed an apparent K_D_ = 0.4 µM and K_D_ = 1.5 µM for *Er*Cry1a and *Er*Cry1b, respectively. 

## 4. Discussion

In this manuscript, we have identified the α-subunit of the cone-specific heterotrimeric G protein as a protein–protein interaction partner of migratory bird *Er*Cry4a. Since G proteins are essential components in many signaling pathways, we suggest that this interaction most likely represents the first step in the biochemical reaction cascade underlying light-dependent magnetoreception in night-migratory songbirds. The present investigation is based on two previous studies showing that recombinant *Er*Cry4a can be purified with a bound FAD chromophore, it exhibits a photo-induced electron transfer pathway, and it is magnetically sensitive in vitro [19]. Using a yeast two-hybrid screening, Wu et al. [22] identified six putative protein interaction partners of *Er*Cry4a, including the α-subunit of the cone-specific G protein. Heterotrimeric G proteins are key proteins in classical signaling cascades [38,39], but a direct connection to a migratory bird cryptochrome has never been observed before. We discuss our findings that support the interaction of *Er*Cry4a with *Er*Gtα being the primary signaling step in magnetoreception.

First, we show a direct binding of *Er*Cry4a and *Er*Gtα on the molecular level and investigated the kinetics of the binding process using SPR. The affinity of myristoylated Gtα/Giα binding to ErCry4a was moderate to high with an apparent K_D_ = 0.29 ± 0.08 µM (e.g., Figure 1A), but the affinity of non-myristoylated Gtα/Giα was about eight-fold higher in the nanomolar range (Figure 1B). Thus, the binding process occurs with moderate to high affinity and is similar to previous findings for the G protein transducin interacting with rhodopsin [40,41,42,43]. The results further showed that the myristoyl group in Gtα/Giα decreased the affinity for *Er*Cry4a. Differences in the observed rate constants point to a plausible explanation for this observation. Dissociation rate constants differ by factors between 9 and 16, but the association rate of *Er*Cry4a to non-myristoylated Gtα/Giα was 70 to 100-fold higher (compare k_a1_ and k_d1_ values in the legend of Figure 1A,B). In a photoreceptor cell, native myristoylated Gtα is fixed to the plane of the membrane by integrating the myristoyl group into the lipid bilayer, leaving the apo-part of the protein freely accessible [44]. However, our experimental setting does not have a lipid platform to anchor Gtα/Giα via a myristoyl group. It would require a completely different experimental design, which is not realizable in our laboratory at present. Due to the lack of an anchoring lipid platform, we assume that the myristoyl group remains rather flexible near the protein surface in the experiments reported here. This orientation could interfere with the association process leading to a lower association rate constant. In the case of non-myristoylated Gtα/Giα, such a barrier did not exist and access of *Er*Cry4a was not hindered, resulting in higher association rates. Our results obtained with non-myristoylated Gtα/Giα resemble, therefore, the cellular situation and indicate a high-affinity interaction with *Er*Cry4a.

Second, we studied the interaction of *Er*Cry4a and *Er*Gtα in a cellular environment using transfection of a neuroretinal quail cell line with fluorescently labeled constructs of *Er*Cry4a and *Er*Gtα in a FRET analysis. The results in Figure 4A,B not only confirm the SPR binding studies employing purified proteins but they also further demonstrate that *Er*Cry4a and *Er*Gtα interact in living cells. Since the *Er*Gtα fluorescence constructs are based on the *Er*Gtα wildtype amino acid sequence and not on the chimeric sequences used for the heterologous expression of *Er*Gtα in *E. coli*, we can also exclude any interferences or false-positive results caused by the insertion in Gtα/Giα. However, we cannot exclude that the interaction kinetics differ in living cells or in vivo. 

Third, how does the binding process of Gtα/Giα to *Er*Cry4a fit into a physiological context? To examine this question, we need to compare our results with the canonical binding of G proteins to opsin receptor molecules. Although no information about binding of European robin G protein to iodopsin or any other opsin is available so far, we refer to information from the well-studied bovine transducin and rhodopsin system. Light-activated rhodopsin shows strong binding to transducin with apparent K_D_ values in the nanomolar to lower micromolar range [40,41,42,43]. However, Dell’Orco and Koch (2011) reported in their SPR study a K_D_ of 0.36 µM for the binding of transducin to dark-adapted rhodopsin [43], which is similar to the affinity that we observed for the binding of myristoylated Gtα/Giα to *Er*Cry4a, but lower than the binding of non-myristoylated Gtα/Giα to *Er*Cry4a. If we assume similar affinities for the interaction of European robin G protein to iodopsin and consider the binding process of *Er*Cry4a and non-myristoylated Gtα/Giα to reflect the cellular situation in a cone photoreceptor cell, the high affinity of the binding process could well compete with dark-adapted iodopsin for binding to *Er*Gtα. However, once iodopsin is activated by light, the binding affinity for the G protein very likely increases as this was observed for the binding of bovine transducin to light-activated rhodopsin reaching K_D_-values in the lower nanomolar range [43].

Fourth, are there structural conditions that agree with previous observations? Wu et al. (2020) also reported that the C-terminus of *Er*Cry4a is not essential for the G_t_α/Giα interaction process [22]. Conducting SPR experiments with a truncated variant of *Er*Cry4a (*Er*Cry4a-497) confirmed this result and showed further that G_t_α/Giα bound to *Er*Cry4a-497 with similar affinity (K_D_ = 0.2 µM, Figure 1C) in comparison to *Er*Cry4a wildtype. Interestingly, non-myristoylated G_t_α/Giα bound to *Er*Cry4a-497 with lower affinity (K_D_ = 0.36 µM) compared to wildtype *Er*Cry4a. For this, we do not have a clear interpretation but point out that purified *Er*Cry4a-497 was less stable compared to the wildtype protein leaving some ambiguities about the quantitative kinetic analysis of this variant. However, similar FRET efficiencies between wildtype Gtα and *Er*Cry4 either with or without the C-terminus in a transiently transfected bird neuroretinal cell line confirm the in vitro data and suggest that these two proteins also interact in living cells and that the C-terminal tail of *Er*Cry4 is not involved in the interaction (Figure 4).

Finally, we extended the interaction analysis of myristoylated G_t_α/Giα to the Cry forms *Er*Cry1a and *Er*Cry1b, which are also expressed in retinal cell types of migratory birds, pointing to putative roles in magnetic sensing [10,16,45,46]. SPR sensorgrams showed robust binding signals in a concentration-dependent manner (Figure 5). Interaction analysis revealed affinities with apparent K_D_ values of 0.4 and 1.53 µM for *Er*Cry1a and *Er*Cry1b, respectively. These values are similar to those determined for *Er*Cry4a and did not disclose a preference of G_t_α/Giα for either of them. However, the shape of the SPR sensorgrams indicates differences in the association and dissociation rate constants, which was confirmed by nonlinear curve fitting (main text and legends of Figure 1 and Figure 5). Association rate and dissociation rates of *Er*Cry4a with G_t_α/Giα were up to ten-fold lower compared to the rates determined for *Er*Cry1a and *Er*Cry1b. One might argue that association rates are different in a cellular environment, when the native myristoylated Gtα attaches to the membrane, allowing diffusion only in a two-dimensional plane and a faster collision with the target. However, once a complex of Gtα with an *Er*Cry variant forms, the dissociation rate is much lower in the case of *Er*Cry4a. In other words, a complex of *Er*Cry4a with G_t_α seems more stable compared to a complex of *Er*Cry1a/b with G_t_α. Taking the rate constants into consideration might explain the apparent discrepancy of the present findings with the yeast two-hybrid experiments by Wu et al. (2020), who found no evidence for *GNAT2* (G_t_α) interacting with *Er*Cry1a [22]. An apparent positive interaction of *Er*Cry1b with G_t_α was observed, but reporter gene expression signals indicating a binding process did not differ from control incubations. Thus, reduced stability of complexes involving *Er*Cry1a and *Er*Cry1b could account for the differences in the yeast two-hybrid experiments. Furthermore, binding of G_t_α/Giα to different *Er*Cry isoforms seems to be a new variation of a common theme since G proteins exhibit a multiplicity of G protein-coupling and targeting [38,39]. For example, transducin binds to the mammalian rod and cone guanylate cyclase type 1 (GC-E) and to glyceraldehyde-3-phosphate dehydrogenase [47,48], two rather different enzymes in different physiological settings. 

## 5. Conclusions

In summary, we provide strong experimental evidence from in vitro data and measurements in living cells showing that *Er*Cry4a and the α-subunit of a cone-specific G protein from European robin interact. We speculate that this interaction could be the first trigger step of biochemical signaling in radical-pair-based magnetoreception. This finding will open several routes of research addressing potential downstream signaling pathways, the sequence of individual binding steps, and the colocalization of signaling components in retinal cells.

## Figures and Tables

**Figure 1 cells-11-02043-f001:**
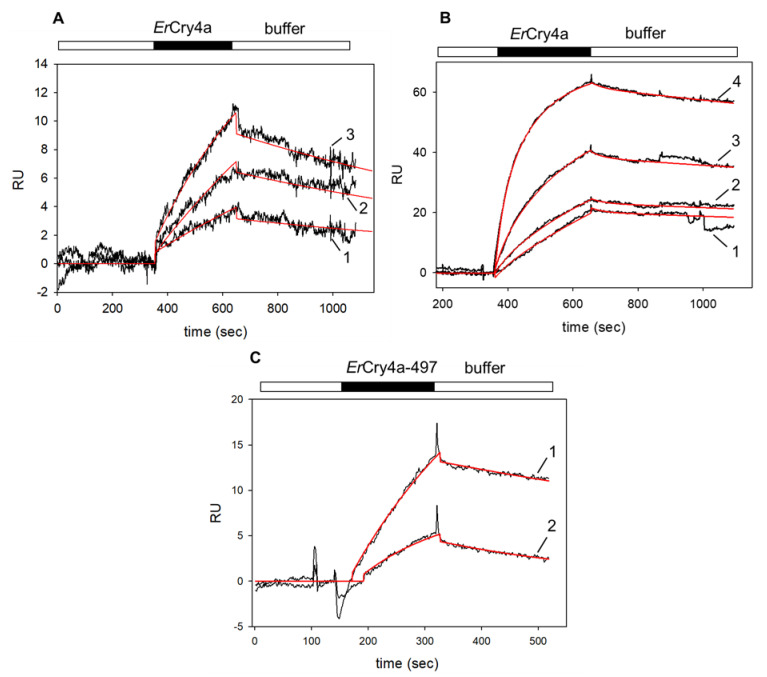
SPR recordings of *Er*Cry4a interacting with the immobilized G_t_α/Giα chimera. Black bars indicate the association phase, open bars indicate buffer flow. For all recordings, we observed that larger RU values resulted from the injection of higher *Er*Cry4a concentrations. (**A**) Sensorgrams obtained after flushing of 50 nM (1), 100 nM (2) and 500 nM (3) *Er*Cry4a over immobilized myristoylated G_t_α/Giα, which led to the formation of an *Er*Cry4a-G_t_α/Giα complex. Global curve fitting (two-state-reaction model, red lines) resulted in an association rate constant k_a1_ = 3.32 × 10^3^ M^−1^ s^−1^ and a dissociation rate constant k_d1_= 6.74 × 10^−4^ s^−1^, K_D_ = 0.20 µM. (**B**) Binding of *Er*Cry4a to non-myristoylated G_t_α/Giα after flushing 5 nM (1), 10 nM (2), 20 nM (3) and 50 nM (4) *Er*Cry4a over immobilized G_t_α/Giα. Global curve fitting (two-state-reaction model, red lines) yielded the following constants: k_a1_ = 3.43 × 10^5^ M^−1^ s^−1^; k_d1_ = 0.011 s^−1^ yielding a K_D_ = 32 nM. (**C**) Injection of 500 nM truncated *Er*Cry4a-497 over myristoylated (1) and non-myristoylated (2) G_t_α/Giα, upper and lower curve, respectively. Curve fitting (red lines) to a Langmuir binding model yielded for the upper sensorgram: k_ass_ = 5.35 × 10^3^ M^−1^ s^−1^; k_diss_ = 9.15 × 10^−4^ s^−1^; for the lower sensorgram: k_ass_ = 6.51 × 10^3^ M^−1^ s^−1^; k_diss_ = 3.05 × 10^−3^ s^−1^.

**Figure 2 cells-11-02043-f002:**
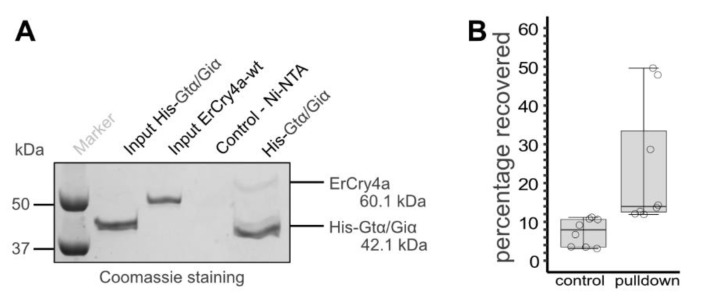
Pulldown experiments of *Er*Cry4a and the His-tagged G_t_α/Giα chimera. (**A**) SDS-PAGE gel of a representative pulldown experiment employing purified proteins to investigate if *Er*Cry4a interacts with the His-tagged G_t_α/Giα chimera. Input lanes show the proteins used for the pulldown experiment. As a negative control, only *Er*Cry4a, but no His-tagged G_t_α/Giα chimera, was incubated with a Ni-NTA affinity matrix. For the experiment, a pre-incubated mixture of both proteins was incubated with the matrix. The full size of the gel image is provided in the Appendix A. (**B**) Densitometry analysis of eight pulldown experiments were carried out. Bars correspond to the average percentage of *Er*Cry4a recovered after the pulldown with error bars, individual results are marked by circles.

**Figure 3 cells-11-02043-f003:**
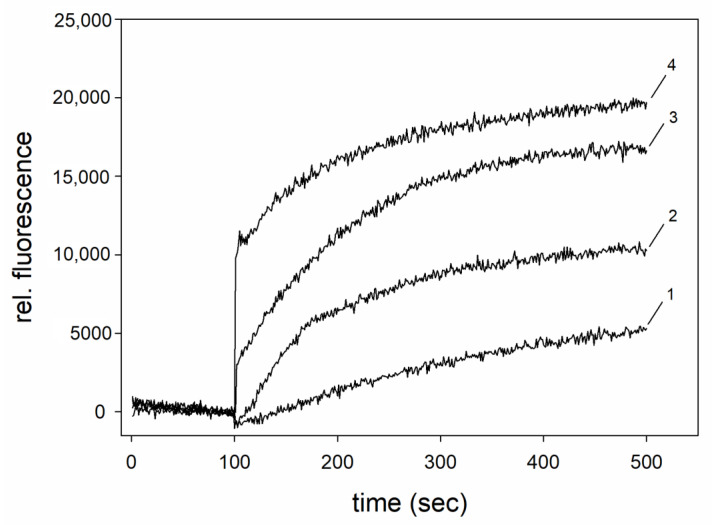
Functional test of the G_t_α/Giα chimera. G_t_α/Giα was present at 1 µM in fluorescence buffer. Trp fluorescence emission recording after injection of 50 µM AlF_4_^−^. Different MgCl_2_ concentrations were present in fluorescence buffer, as indicated, and resulted in a successive increase in relative fluorescence emission: (1) zero extra addition of MgCl_2_ in buffer, very small amount of MgCl_2_ present after the purification procedure; (2) 100 µM (2); 500 µM (3); 1000 µM (4).

**Figure 4 cells-11-02043-f004:**
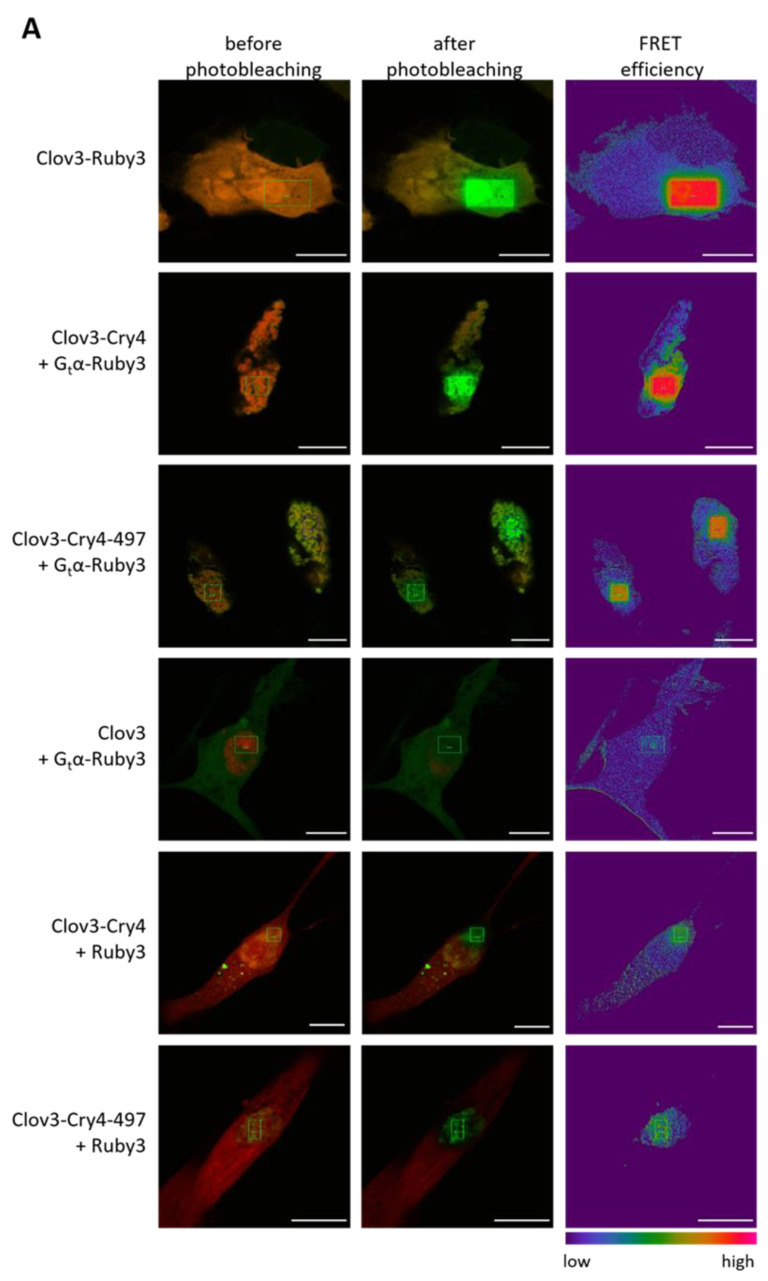
Interaction between G_t_α and *Er*Cry4a with and w/o C-terminus (*Er*Cry4-497) using the acceptor photobleaching FRET technique. (**A**) Representative confocal laser scanning microscopy images of QNR/K2 cells before and after photobleaching (overlay of red-donor and green-acceptor), and FRET efficiencies. Expressed proteins as indicated. The green rectangular indicates the bleached area. Scale bar is 10 µm. (**B**) FRET efficiencies as determined by the LAS AF software between N-terminally tagged Cry4 (or the truncated version Cry4-497) with mClover3 (Clov3) and G_t_α, C-terminally tagged with mRuby3 (Ruby3), including negative and positive controls. Boxplot graphs were generated using Origin graph software. The boxes range from Q1 (the first quartile) to Q3 (the third quartile) of the data distributions, and the range represents the IQR (interquartile range). Medians are indicated by lines across the boxes. The whiskers extend between the most extreme data points, excluding outliers, which are defined as being outside 1.5 × IQR above the upper quartile and below the lower quartile. Each data point represents the measurement for one cell, with n indicating total cell numbers measured in each condition. Each condition was tested in at least three separate experiments.

**Figure 5 cells-11-02043-f005:**
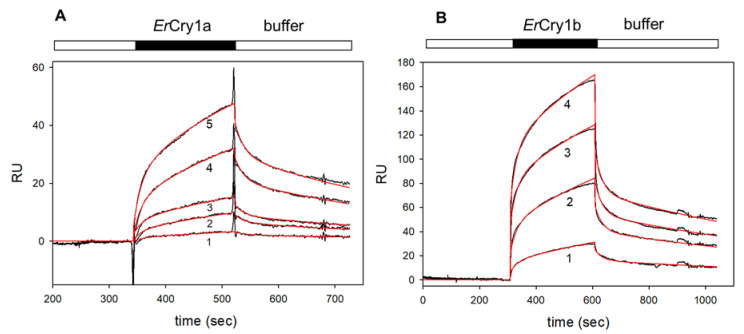
SPR recordings of *Er*Cry1a and *Er*Cry1b interacting with immobilized myristoylated G_t_α/Giα. (**A**) Injection of increasing concentrations of *Er*Cry1a, 25 nM (1), 35 nM (2), 50 nM (3), 70 nM (4) and 100 nM (5) yielded increasing amplitudes (sensorgrams 1–5). Global curve fitting using a two-state-reaction model gave a forward reaction rate k_a1_ = 1.3 × 10^5^ M^−1^s^−1^ and a backward reaction rate k_d1_ = 5.27 × 10^−2^ s^−1^, K_D_ = 0.4 µM. (**B**) Injection of increasing concentrations of *Er*Cry1b, 50 nM (1), 150 nM (2), 300 nM (3) and 400 nM (4) yielded increasing amplitudes (sensorgrams 1–4). Global curve fitting using a two-state-reaction model gave a forward reaction rate k_a1_ = 2.17 × 10^4^ M^−1^s^−1^ and a backward reaction rate k_d1_ = 3.32 × 10^−2^ s^−1^, K_D_ = 1.53 µM.

**Table 1 cells-11-02043-t001:** Overview of FRET plasmids and expressed proteins.

Plasmids	Expressed Protein
pKan-CMV-mClover3	mClover3
pKan-CMV-mRuby3	mRuby3
pKan-CMV-mClover3-mRuby3	mClover3-mRuby3
pKan-CMV-mClover3-*ErCRY*4	mClover3-*Er*Cry4
pKan-CMV-mClover3-*ErCRY*4-497	mClover3-*Er*Cry4, truncated after aa 497
pKan-CMV-*ErGNAT2*-mRuby3	*Er*G_t_α-mRuby3

**Table 2 cells-11-02043-t002:** Statistics of FRET monitoring. ANOVA results of the binomial GLM for differences between the interactions Cry4 and Cry4-497 with Gtα and the respective negative controls with mRuby3 or mClover3 only. The respective sample sizes (*N*), resulting chi-squared statistics (*χ*2) and *p*-values are listed for each comparison. Significant differences are indicated with asterisks, corresponding *p*-values are in bold (*p* ≤ 0.05 = *; *p* ≤ 0.01 = **; *p* ≤ 0.001 = ***.

	*N*		*N*	*χ*2	*p*	
mClover3-Cry4 + Gtα-mRuby3	25	mClover3 + Gtα-mRuby3	31	9.854	**0.0017**	**
		mClover3-Cry4 + mRuby3	24	7.100	**0.0077**	**
mClover3-Cry4-497 + Gtα-mRuby3	45	mClover3 + Gtα-mRuby3	31	12.525	**0.0004**	***
		mClover3-Cry4-497 + mRuby3	39	6.072	**0.0137**	*

## Data Availability

Accession numbers of genes are KT380948.1 (ErCry1a), KT380949.1 (ErCry1b), KX890129.1 (ErCry4a). All data are available in the main text and/or the Appendix A. Additional data to this paper may be requested from the authors.

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
