# Peer review of "Direct Interaction of Avian Cryptochrome 4 with a Cone Specific G-Protein"

_cells, 2022, doi:10.3390/cells11132043_

Round 1

Reviewer 1 Report

The manuscript by Görtemaker et al. describes for the first time the interaction of a cone G protein quimera with avian cryptochrome 4. From the analysis of their results, both in vitro and in a cellular system, the authors propose that this might be the first biochemical step in radical-pair-based magnetoreception in birds.

On the formal side, the manuscript is well organized in relevant sections, the methods described to a good level of detail (with the addition of useful methodological information in the supplementry material file) and the figures of a good quality.

This is an experimental study conducted to a high scientific standard using a combination of biochemical methods (including cloning of genes and the purification of proteins and pull-down assays) and biophysical methods to probe the interaction between the G protein and the receptor (SPR and FRET). All the experiments have been carefully conducted and the appropiate controls are included to rule out misinterpretation of the results. The results are also analyzed for statistical significance and discussed in detail in the Discussion section. Overall the manuscript is well written and the conclusions derived are backed up by the results presented.

There are a few methodological aspects that the authors could further consider in their discussion or that they could help clarify.

1) First of all a nomenclature aspect. The authors indicate that their study in a cellular model system is in vivo. This reviewer has some doubts if a cell culture system can be refered to as in vivo or this terminology is reserved for work with a living system (i.e. animal models).

2) One concern is the use of a chimeric G protein. The authors have to use this strategy to be able to successfully purify this protein. On the other side, this is an approach that has been used but there may be some issues concerning the extrapolation of the results obtained with the chimeric G protein to the in vivo system. Is it possible that this difference can potentially affect the in vivo situation. The authors should further elaborate on that.

3) Another concern is the difference in the interaction profiles (affinities) concerning the myristoylated and non-myristoylated G protein chimera. This difference was the contrary as expected in the SPR profiles. In any event, the authors provide a good explanation in the discussion section to account for this observed effect. However, the authors could further emphasize on the effect of the lipid system in their results. It could be certainly interesting to experimentally analyze the effect of lipids on their results.

4) Figure 3 convincingly shows the functional in vitro activation of the chimeric G protein. But, what would be the meaning of the sudden increase of the signal in the fluorescence curves at high Mg2+ concentrations?

5) In the discussion section, it is stated that "no information regarding binding of European robin G protein to iodopsin or any other opsin is available so far". The authors then compare their results to the vertebrate opsin-transducin interaction. This is a legitimate and appropriate comparison. But, would the authors expect that useful information could be obtained from the combined use of some of the components of the vertebrate and the avian systems to explore further the mechanistic details of such interactions? This aspect could be comented by the authors.

6) Labeling of Figure 4. Negativ controls should be negative controls.

Author Response

Dear Reviewer,

we thank you for the helpful comments and provided a point-by-point response.

Best regards,

Karl-W. Koch

------

Response to reviewer comments

Comments and Suggestions for Authors

The manuscript by Görtemaker et al. describes for the first time the interaction of a cone G protein quimera with avian cryptochrome 4. From the analysis of their results, both in vitro and in a cellular system, the authors propose that this might be the first biochemical step in radical-pair-based magnetoreception in birds.

On the formal side, the manuscript is well organized in relevant sections, the methods described to a good level of detail (with the addition of useful methodological information in the supplementry material file) and the figures of a good quality.

This is an experimental study conducted to a high scientific standard using a combination of biochemical methods (including cloning of genes and the purification of proteins and pull-down assays) and biophysical methods to probe the interaction between the G protein and the receptor (SPR and FRET). All the experiments have been carefully conducted and the appropiate controls are included to rule out misinterpretation of the results. The results are also analyzed for statistical significance and discussed in detail in the Discussion section. Overall the manuscript is well written and the conclusions derived are backed up by the results presented.

RE: We thank the referee for the encouraging and helpful comments and address all points in the sections below.

There are a few methodological aspects that the authors could further consider in their discussion or that they could help clarify.

1) First of all a nomenclature aspect. The authors indicate that their study in a cellular model system is in vivo. This reviewer has some doubts if a cell culture system can be refered to as in vivo or this terminology is reserved for work with a living system (i.e. animal models).

RE: We agree with the reviewer that the terminology might be misleading. We changed “in vivo” to “in living cells”, which should be appropriate.

2) One concern is the use of a chimeric G protein. The authors have to use this strategy to be able to successfully purify this protein. On the other side, this is an approach that has been used but there may be some issues concerning the extrapolation of the results obtained with the chimeric G protein to the in vivo system. Is it possible that this difference can potentially affect the in vivo situation. The authors should further elaborate on that.

RE: We cannot exclude that the in vivo situation (in the retina of a living bird) differs in certain aspects from our results obtained in vitro and in cell culture. However, the vast experience obtained over decades from investigation of G protein-triggered phototransduction tells us that fundamental discoveries of the basics in phototransduction that were made originally in vitro were later well reproduced by in vivo studies (using transgenic mice). Any comments made regarding this aspect for magnetoreception is however highly speculative at present. In any case, we were fully aware of the possible drawbacks caused by the use of the chimeric mutant and stated this already in the paragraph about the FRET measurements:

“Next, we wanted to verify that ErCry4a interacts in a cellular environment with the wildtype form of ErGtα. This is important to exclude that the observed interactions were not compromised by the use of a chimeric construct.”

In the revised version submitted, we further added a short statement at the end of the third paragraph of the Discussion part reiterating this.

3) Another concern is the difference in the interaction profiles (affinities) concerning the myristoylated and non-myristoylated G protein chimera. This difference was the contrary as expected in the SPR profiles. In any event, the authors provide a good explanation in the discussion section to account for this observed effect. However, the authors could further emphasize on the effect of the lipid system in their results. It could be certainly interesting to experimentally analyze the effect of lipids on their results.

RE: We agree with the reviewer that the effect of membranes on the interaction profiles is of high relevance, but a comprehensive SPR study would involve the immobilization of lipid bilayers on chip surfaces and the investigation of several interaction modes including Cry4a-lipid, G protein-lipid (myristoylated and non-myristoylated), Cry4a-G protein in the presence and absence of membranes. Additional variations are the lipid composition and other Cry variants. This is therefore far beyond the content of the present manuscript. We are currently considering how we can perform such a study in the future.

4) Figure 3 convincingly shows the functional in vitro activation of the chimeric G protein. But, what would be the meaning of the sudden increase of the signal in the fluorescence curves at high Mg2+ concentrations?

RE: The sudden increase was not surprising to us, because it is already visible at lower Mg2+-concentrations. It matches the concentration dependency that is observed in Figure 3 and indicates the immediate monitoring of the conformational change, which occurs with a delay at lower Mg2+-concentrations due to limited diffusion.

5) In the discussion section, it is stated that "no information regarding binding of European robin G protein to iodopsin or any other opsin is available so far". The authors then compare their results to the vertebrate opsin-transducin interaction. This is a legitimate and appropriate comparison. But, would the authors expect that useful information could be obtained from the combined use of some of the components of the vertebrate and the avian systems to explore further the mechanistic details of such interactions? This aspect could be comented by the authors.

RE: We agree with the suggestion of the reviewer that useful information might be obtained from the combined use of the vertebrate and avian system. Our future approach will, however, focus on investigating a pure avian system. These studies are in currently progress but far from finished.

6) Labeling of Figure 4. Negativ controls should be negative controls.

RE: done 

Reviewer 2 Report

This manuscript describes a protein-protein interaction between cryptochrome 4 (ErCry4a) and Gtα derived from European robin, a night-migratory bird. The study used three independent methods to carry out this; pulldown and SPR in vitro and FRET in vivo. The methods are described very well and the results are largely convincing. This is a part of a larger research project to elucidate how migratory birds sense the Earth´s magnetic field via cryptochrome 4a. The study demonstrated the interaction between the two proteins convincingly. However, how this interaction is responsible, or a part of the magnetic field sensing is not clear and highly speculative. The discussion/interpretation of the results derived from myristoylated and non-myristoylated Gtα is not convincing. SPR experiments could have included membrane components to address the issues related to myristoylation. One obvious question is why not use an active form of Gtα in the binging experiments.

Here are some other concerns and minor issues:

L24, should be “European robin cryptochrome 4a (ErCry4a)”

L234-260, the myristoylation state of the G protein should be included.

L306, “black diamonds” should be “circles.”

L336-342, The content of this paragraph seems to have no relevance to the manuscript.

L406-414, the text should be more accurate. For example, text like “Gtα/Giα chimera was used as a surrogate for Gtα.” can be included.

 Figure 2A, The position of the band of ErCry4a in pulldowned sample seems higher than the band of input ErCry4a. This is more evident in Figure S6. This should be explained. The nature of the band should have been identified by another method (i.e., Western blotting).

L526, “activity tests” seems out of place.

L539, “Clov3-Gtα-Ruby3” should be “Clov3+Gtα-Ruby3.”

L607-610, the text seems to be too speculative. There is no experimental evidence presented to demonstrate that this is “the first” protein-protein interaction or “the first” step, considering their previous paper demonstrating 5 other proteins interacting with ErCry4a.

L633-638, is highly speculative.

L706-707, same comment as L607-L610.

Author Response

Dear Reviewer,

thank you very much for the helpful comments. We addressed all comments by a point-by-point response.

Kind regards,

Karl-W. Koch

-------

Response to reviewer comments

Review 2

Comments and Suggestions for Authors

This manuscript describes a protein-protein interaction between cryptochrome 4 (ErCry4a) and Gtα derived from European robin, a night-migratory bird. The study used three independent methods to carry out this; pulldown and SPR in vitro and FRET in vivo. The methods are described very well and the results are largely convincing. This is a part of a larger research project to elucidate how migratory birds sense the Earth´s magnetic field via cryptochrome 4a. The study demonstrated the interaction between the two proteins convincingly. However, how this interaction is responsible, or a part of the magnetic field sensing is not clear and highly speculative. The discussion/interpretation of the results derived from myristoylated and non-myristoylated Gtα is not convincing. SPR experiments could have included membrane components to address the issues related to myristoylation. One obvious question is why not use an active form of Gtα in the binging experiments.

RE: We thank the referee for the encouraging comments and address all points in the sections below. Concerning the use of lipid membranes in SPR studies, we refer to our response to point 3) of referee 1. Further, we would like to point out that we used an active form of Gtα (see tryptophan-fluorescence measurements in Figure 3). The reason, why we used a chimeric form of Gtα is explained on page 3, L121-124. 

Here are some other concerns and minor issues:

L24, should be “European robin cryptochrome 4a (ErCry4a)”

RE: done

L234-260, the myristoylation state of the G protein should be included.

RE: addition added to paragraph “Expression and purification of the myristoylated Gtα/Giα chimera”

L306, “black diamonds” should be “circles.”

RE: done

L336-342, The content of this paragraph seems to have no relevance to the manuscript.

RE: we keep this short paragraph because it relates to the construction of the cDNA library that we used.

L406-414, the text should be more accurate. For example, text like “Gtα/Giα chimera was used as a surrogate for Gtα.” can be included.

RE: done

Figure 2A, The position of the band of ErCry4a in pulldowned sample seems higher than the band of input ErCry4a. This is more evident in Figure S6. This should be explained. The nature of the band should have been identified by another method (i.e., Western blotting).

RE: Indeed there is a minor shift in the observed electrophoretic mobility observed for ErCry4a comparing input to pulldown sample. We assume that this is due to differences in the electric field across the gel and thus does not impact the validity of our results. Importantly, the assay was carried out using purified proteins - and not cell lysates - as demonstrated by the input lanes. Thereby we can exclude that another protein except ErCry4a has been pulled down by the immobilized His-Gtα/Gia. To make this clearer, we adapted the figure texts of Figure 2 and S6 to highlight that purified components have been used for the pulldown:

... SDS-PAGE gel of a representative pulldown experiment employing purified proteins to investigate ....

L526, “activity tests” seems out of place.

RE: we deleted “activity”

L539, “Clov3-Gtα-Ruby3” should be “Clov3+Gtα-Ruby3.”

RE: done

L607-610, the text seems to be too speculative. There is no experimental evidence presented to demonstrate that this is “the first” protein-protein interaction or “the first” step, considering their previous paper demonstrating 5 other proteins interacting with ErCry4a.

RE: We toned down the wording. However, please note that the last sentence in the paragraph is already stating: ”We discuss our findings that support the interaction of ErCry4a with ErGtα being the primary signaling step in magnetoreception.” We consider a discussion of results in a putative physiological framework as a legitimate task of a manuscript.

L633-638, is highly speculative.

RE: We toned down the wording.

L706-707, same comment as L607-L610.

RE: We modified the conclusion by changing “in vivo” to “in living cells” and by clearly separating what we show and what we suggest. Please see also our response to referee 1, point 1).